# 3D Foot Reconstruction Based on Mobile Phone Photographing

**Lulu Niu** [1,2], **Gang Xiong** [2,3], **Xiuqin Shang** [4], **Chao Guo** [2], **Xi Chen** [5] and **Huaiyu Wu** [2,*]

1   The School of Artificial Intelligence, University of Chinese Academy of Sciences, Beijing 100049, China; niululu2018@ia.ac.cn
2   The State Key Laboratory for Management and Control of Complex Systems, Institute of Automation, Chinese Academy of Sciences, Beijing 100190, China; gang.xiong@ia.ac.cn (G.X.); guochao2014@ia.ac.cn (C.G.)
3   The Guangdong Engineering Research Center of 3D Printing and Intelligent Manufacturing, The Cloud Computing Center, Chinese Academy of Sciences, Dongguan 523808, China
4   The Beijing Engineering Research Center of Intelligent Systems and Technology, Institute of Automation, Chinese Academy of Sciences, Beijing 100190, China; xiuqin.shang@ia.ac.cn
5   Department of Electrical and Information Technology, LTH Box 118, SE-221 00 Lund, Sweden; xi7224ch-s@student.lu.se
*   Correspondence: huaiyu.wu@ia.ac.cn

**Abstract:** Foot measurement is necessary for personalized customization. Nowadays, people usually obtain their foot size by using a ruler or foot scanner. However, there are some disadvantages to this, namely, large measurement error and variance when using rulers, and high price and poor convenience when using a foot scanner. To tackle these problems, we obtain foot parameters by 3D foot reconstruction based on mobile phone photography. Firstly, foot images are taken by a mobile phone. Secondly, the SFM (Structure-from-Motion) algorithm is used to acquire the corresponding parameters and then to calculate the camera position to construct the sparse model. Thirdly, the PMVS (Patch-based Multi View System) is adopted to build a dense model. Finally, the Meshlab is used to process and measure the foot model. The result shows that the experimental error of the 3D foot reconstruction method is around 1mm, which is tolerable for applications such as shoe tree customization. The experiment proves that the method can construct the 3D foot model efficiently and easily. This technology has broad application prospects in the fields of shoe size recommendation, high-end customized shoes and medical correction.

**Keywords:** 3D foot reconstruction; SFM; PMVS; personalized customization





## 1. Introduction

In the era of Industry 4.0, consumers are paying more and more attention to the demand for personalized products. With the cooperation of E-commerce and mobile terminals, a variety of commodities have emerged in APP stores, resulting in an increasing demand for users to carry out simple and feasible model reconstructions of real scenes and objects. Shoes and clothing, medical treatment and other industries are collecting 3D big data of the human body to achieve high-end personalized product manufacturing [1]. Shoe manufacturers also hope to design and produce shoes that accord with the physiological characteristics of the human body to make their customers more comfortable [2]. More and more, users hope to obtain a 3D foot model in a convenient way without leaving home. To achieve this goal, the first step is to achieve fast and automatic measurement of foot parameters.

Accurately measuring users' data has become an exigent and intractable subject for high-end customization. At present, most companies still use backward traditional contact manual measurement methods. The traditional foot data collection methods can be divided into two sorts. The first is manual measurement as shown in Figure 1a,b. Using a cloth ruler, steel ruler and other tools may lead to a deviation due to the errors in the measurement position or technique. Besides, the work efficiency and the repetition rate are relatively

low, and the error between different surveyors is large. The second is the foot scanner, which uses laser scanning—the laser lines are emitted from one or more laser sources at a certain frequency. Multiple cameras are used to collect multiple frames of images from different angles to obtain foot data. However, the equipment is expensive, bulky, has poor portability and high maintenance cost in the later stages. To tackle the problems of large measurement error manual measurement and the poor convenience of foot scanners, this paper proposes a 3D model reconstruction method based on mobile phone photographs.

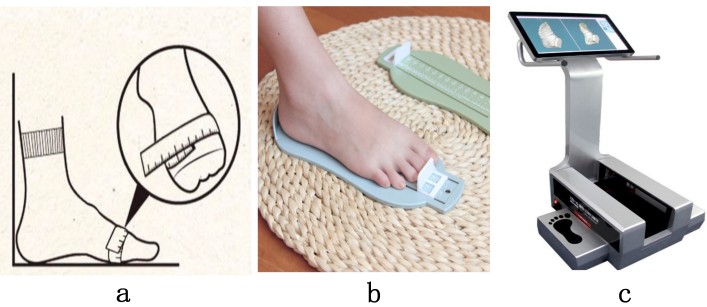

**Figure 1.** Foot data acquisition method. (**a**) Ruler; (**b**) Foot measuring device; (**c**) Foot scanner.

Foot 3D reconstruction is different from other objects, because the overall volume of the foot is small, and the surface of the foot is uneven. Besides, the texture feature of foot skin is not obvious, which makes it difficult to establish the corresponding relationship of the same point in different images, leading to an unsatisfactory reconstruction effect. Therefore, in our experiment, the volunteer wears colored socks and stands on a magazine, so as to obtain rich texture features and distinguish the surrounding environment for better reconstruction effect. On the other hand, magazines are used as a reference for better calculation of 3D foot parameters.

The method allows users to take more multi-view photos of the foot with a mobile phone, preferably overlapping areas, and then the images reconstruct the 3D foot model with the relevant algorithm. The experimental error of the 3D foot reconstruction method is around 1mm compared with the manual measurement, which is close to the scanning result of the foot scanner and is also tolerable for applications such as shoe tree customization.

The follow-up structure of the paper is as follows—Section 2 introduces the existing techniques and methods of 3D reconstruction; Section 3 describes the principle and algorithm flow of the 3D model reconstruction of the foot based on mobile phone photography; Section 4 provides experimental results and analysis; Section 5 summarizes the full text and analyzes future development trends.

## 2. Related Works

Three dimensional computer vision realizes the perception, recognition, and understanding of 3D scenes in the objective world with a computer. Based on the 3D reconstruction technology of vision, the camera is used to obtain the image or video of the object from multiple views. According to the projection relationship between the object and the image and the matching relationship between the images under multiple-view, the object is reconstructed by combining the position and shape of the object and other information. Three dimensional computer vision systems include camera calibration, features extraction, stereo matching and 3D reconstruction [3].

Since its inception, 3D reconstruction has attracted countless scholars to continuously expand and improve it. Faugeras proposes a 3D model reconstruction method based on hierarchical reconstruction [4]. With the improvement of computing performance, a 3D reconstruction method of dense point clouds appeared; with dense matching [5] of stereo image pairs based on a region growth algorithm, the result of the reconstruction is close to the real model. This paper focuses on the 3D reconstruction method based on an image sequence. Since the images are all two-dimensional, it is difficult to reconstruct the 3D

structure of the target object if only one camera is used to capture a still picture, so we must move the camera or the object, which is called the recovery SFM [6]. Originally, in 1981, Longuet-Higgins first proposed an algorithm for reconstructing scenes [7] from eight corresponding points or more in two images, published in *Nature*. The concept of the essential matrix is introduced in the algorithm. After that, Hartly et al. [8] found that the essential matrix can also be applied to the case where the two views are not calibrated, but the result of the 3D reconstruction is different from the real scene by a projection transformation, and thus they further obtained the basic matrix. This matrix completely depicts the projection relationship between the two views and later, Zhengyou Zhang [9] used the basic matrix, spatial points and line primitives to perform 3D reconstruction on the basis of predecessors. Mingtao Pei studied the monocular 3D reconstruction based on pure translation based on the position of the parameters in the camera [10]. By adjusting the internal parameters, the 3D reconstruction results of different shapes could be obtained. Ali et al. [11] deduced that a set of relative camera attitudes could be calculated based on the three vertical or parallel edges detected in the image sequence. From this attitude, the image is then reconstructed by depth map calculation, fusion and 3D point cloud restoration. Others use the front background color probability model of the image set and the contour projection for 3D reconstruction [12] in the case of an object reference model. However, these methods are mostly based on the large scene and reconstruct the big object, and the error is large. It does not apply to the reconstruction of small objects such as the foot, because the feature points on the feet are relatively smaller. It is difficult to accurately calculate the foot model using the conventional method of visual reconstruction.

In recent years, other constraints have been added to reduce the errors in the details of the reconstructed model and improve the real-time performance of the optimization algorithm. Brian and Steven's team conducted classic research, making a series of outdoor large buildings reconstructions, including minimizing the energy function smooth depth map [13] based on the PMVS, and dense 3D point clouds using Poisson reconstruction after the grid model, after a series of indoor images combined with ua ser interaction information change model [14]. Since Hans-Andrea introduced the factor graph model in [15], researchers combined the factor graph with the multi-view geometric to build the overall error function, such as Vadim et al. [16] combination of IMU camera parameters filtering, visual projection, feature point matching constraints to build a factor graph by minimizing the global error function calculation for more accurate camera parameters. In recent years, in addition to the use of laser and structural light scanning measurements, more representative algorithms are based on special markers [17] and the 3D foot measurement system is based on binocular stereo matching.

However, these existing methods require a large and expensive foot shape data acquisition machine, which is convenient for camera calibration and it is difficult to generalize to ordinary people. Nowadays, the major mobile intelligent terminal manufacturers have added sensor technology to their products, which provides feasibility for mobile terminals to collect data. So our method is based on the two-dimensional image taken by the mobile phone, then uses the SFM to construct the sparse model and uses the PMVS for a dense model. The experiment proves that the method can construct the 3D model of the foot easily and efficiently.

## 3. 3D Foot Reconstruction Method

The core research tasks of 3D computer vision are scene structure and camera pose, as well as camera parameters. Based on the principle of the camera calibration, the SFM is used for sparse reconstruction. In the process of sparse reconstruction, after taking multiple images of the foot, we used the Scale Invariant Feature Transform (SIFT) algorithm and the Random Sample Consensus (RANSAC) algorithm, estimating the 3D coordinates of the feature points pair and camera parameters, and used the Bundle Adjustment. After that, we used the PMVS [18] to perform dense point cloud reconstruction. After a series of post-

processing processes, we obtained the foot 3D model. The overall experimental process is shown in Figure 2. The following is an introduction to the method used in the paper.

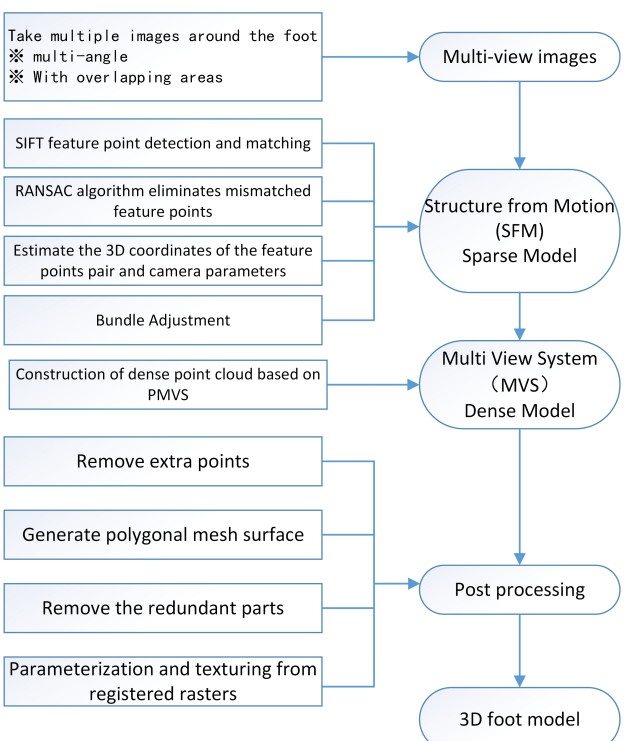

**Figure 2.** Overall experimental process.

### 3.1. Fundamentals of Camera Imaging and Camera Calibration

The camera imaging model is determined by the relationship between each point in the photo and the corresponding point in real space. Since the size of the actual object space is converted to the size of the camera image, camera calibration is required, and the lens imaging process will cause some image distortion, which is called camera distortion. Camera calibration is needed to correct the distortion. The camera calibration is the process of determining the external and internal parameters of the camera. The objects known to the three-dimensional position calibration and their images in the camera image plane (it is usually in the form of a known 3D coordinates calibration point and the image pixel coordinates of these points in the camera image) can be defined. The internal and external parameters of the camera involved in the transformation from a world coordinate system to image pixel coordinate system are solved by the optimization method.

In the imaging process of the camera, the image pixel coordinate system, the camera coordinate system and the world coordinate system are involved. The hypothesis is that the point of the 3D world coordinate system is $P = [X, Y, Z, 1]^T$, and the two-dimensional plane camera coordinate system $m = (u, v, 1)^T$. According to Zhang's Camera Calibration Method [19,20], we assumed that $Z = 0$ of the checkerboard plane; the homography matrix $H$ of the image plane would be solved by the coordinates of point P and point m, and the internal parameters and external parameters of the camera would be obtained. The basic principle is as follows:

$$s \begin{bmatrix} u \\ v \\ 1 \end{bmatrix} = K \begin{bmatrix} r_1 & r_2 & r_3 & t \end{bmatrix} \begin{bmatrix} X \\ Y \\ 0 \\ 1 \end{bmatrix} = K \begin{bmatrix} X \\ Y \\ 1 \end{bmatrix} \tag{1}$$

$$H = [h_1 h_2 h_3] = \lambda K[r_1 r_2 t], \tag{2}$$

where $s$ is the scale factor, $K$ is the camera's internal parameters, $r_1$, $r_2$ are the vectors in the rotation matrix R, and T is the translation vector. According to the nature of the rotation matrix, that is, $r_1^T r_2 = 0$ and $\|r_1\| = \|r_2\| = 1$, the following two basic constraints on the internal parameter matrix can be obtained for each image:

$$h_1^T k^{-T} k^{-1} h_2 = 0 \tag{3}$$

$$h_1^T k^{-T} k^{-1} h_1 = h_2^T k^{-T} k^{-1} h_2. \tag{4}$$

Since the camera has 5 unknown parameters, when the number of captured images is greater than or equal to 3, the $K$ can be solved to obtain the internal parameter matrix and can be obtained from:

$$\lambda = \frac{1}{s} = \frac{1}{\|A^{-1} h_1\|} = \frac{1}{\|A^{-1} h_2\|} \tag{5}$$

$$r_1 = \frac{1}{\lambda} K^{-1} h_1 \tag{6}$$

$$r_2 = \frac{1}{\lambda} K^{-1} h_2 \tag{7}$$

$$r_3 = r_1 \times r_2 \tag{8}$$

$$t = \lambda K^{-1} h_3. \tag{9}$$

After obtaining $K$, $R$, $t$, the spatial point X is calculated by triangulation.

### 3.2. Sparse Reconstruction

Sparse reconstruction uses the SIFT algorithm to extract the image features, and RANSAC is used to eliminate the mismatched feature points. After such iterations, the SFM sparse point cloud image is preliminarily generated and then performs Bundle Adjustment to minimize the reconstruction error. Finally, the SFM sparse point cloud is obtained.

### 3.2.1. Feature Point Detection and Matching

The SIFT [21,22] is a feature detection algorithm in computer vision to detect and describe local features in images. It can be decomposed into the following four steps—detection of scale-space extrema, accurate keypoint licalization, orientation assignment and the local image descriptor [23].

Since the SIFT feature is based on the point of some local appearance on the object, it is independent of the size and the rotation of the images, and they have a high tolerance to light, noise and some microscopic angle changes.

The RANSAC algorithm is a learning technique for estimating the parameters of a model by random sampling of observed data [24]. Given a dataset whose data elements contain both inliers and outliers, RANSAC uses the voting scheme to find the optimal fitting result.

An advantage of RANSAC is its ability to carry out robust estimation [25] of the model parameters, that is, it can estimate the parameters with a high degree of accuracy even when a significant number of outliers are present in the data set. So we use the RANSAC algorithm to eliminate mismatched feature points.

### 3.2.2. Estimating the 3D Coordinates of the Feature Points Pair and Camera Parameters

At present, only the existing foot photo is a known quantity, that is, only the pixel position (x, y) of the feature point to be measured in multiple images is known. Therefore, the incremental SFM method is used for sparse reconstruction. Using the Zhang's Camera Calibration Method mentioned above to directly solve the camera's internal parameter matrix $K$, the camera rotation matrix $R$, the relative translation of the world coordinate t, and the coordinate of the feature point to be measured in the world coordinate system

P(X, Y, Z), then the scale factor $\lambda$ is determined using a reference of a known specification, thereby obtaining the true spatial position coordinates (X, Y, Z) of the feature point.

As shown in Figure 3 above, Image1 and Image2 are randomly selected in multiple images of different angles to determine the initial image pair. The polar geometric constraints of the two views can be described by a $3 \times 3$ matrix, which is called the basic matrix F. F expresses the mapping relationship of the polar line between the point $X_1$ in Image1 and the point $X_2$ in Image2. We can use the 8-points method to find F, and use Zhang's Camera Calibration Method to calculate the initial value $[R \mid t]$ matrix of the internal and external parameters of the camera that captures images Image1 and Image2:

$$\mathbf{x}_1^T \mathbf{F} \mathbf{x}_2 = 0 \rightarrow \mathbf{x}_1^T \mathbf{K}_1^{-T} \mathbf{R}[\mathbf{t}]_\mathbf{x} \mathbf{K}_2^{-1} \mathbf{x}_2 = 0. \tag{10}$$

when the parameter matrices $K_1$ and $K_2$ are known, E = R[t]$_x$ is called an essential matrix, and the essential matrices E1 and E2 corresponding to the Image1 and Image2 are respectively calculated by the 5-point method. Since $E = [R \mid t]$, the rotation matrices $R_1$ and $R_2$ and the translation matrices $t_1$ and $t_2$ with relative to world coordinates can be decomposed from the essential matrix E. Then, the position information of the feature points to be measured in each image is extracted to construct the initial sparse modeling.

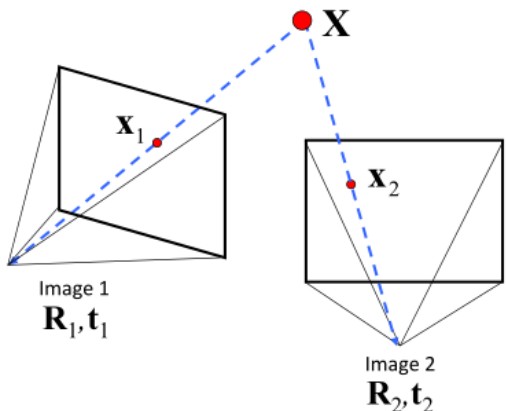

**Figure 3.** Two view geometry.

The initial sparse model is obtained by the above, using the triangulation method to calculate the positions $\lambda(X_1, Y_1, Z_1)$ and $\lambda(X_2, Y_2, Z_2)$ of the feature points to be measured in the world coordinates of the images Image1 and Image2. Inputting Image3, the pixel position of the feature point to be measured in the camera coordinates in the initial sparse modeling, and can reacquire the camera internal and external parameter $[R \mid t]$ matrix, that is, the camera rotation matrix $R_3$ and the translation amount $t_3$ with respect to the world coordinates, and using the internal and external parameters of the camera to modify the initial sparse model. Iteratively, and the position coordinates of the feature points obtained by the above method are corrected by the BA (Bundle Adjustment) method; BA is an optimization model, whose essence is to minimize the reprojection error. BA boils down to minimizing the reprojection error between the image locations of observed and predicted image points, which is expressed as the sum of squares of a large number of nonlinear, real-valued functions [26]. BA can not only optimize the pose (*R* and *t*), but also the spatial position of feature points.

### 3.3. Dense Reconstruction

The dense reconstruction is performed by using the PMVS algorithm. The purpose of the PMVS [18] algorithm is to ensure that at least one patch is projected on each image block [27]. The first step of the PMVS is initializing the feature matching. The 3D surface model reconstruction based on the patch turns the sparse point of the 3D space into a directional seed patch set, and the second step is expanding the patch to generate a dense

patch model, and the third step is the patch filtering, which further filters the target model to generate an accurate 3D model [28].

## 4. Experiment

Experimental equipment and environmental requirements—the mobile phone had a camera resolution of 1920 × 1080, photographing in the natural light. The magazine was used as a calibration object and its size was 210 mm × 297 mm, it was reconstructed with the foot to calculate the foot size. A computer with Intel® Core™ i7-9700 CPU @ 3.00 Ghz (8 CPUs) was used, and the software used was Visual SFM [29] 64-bit, Meshlab_64bit V1.3.3.

### 4.1. Image Data Acquisition

In this experiment, the volunteer wore striped sock stands on the magazine, we used the mobile phone to take a video around the feet, then used Free Studio to capture a photo every second. Finally, we obtained 84 images. Part of the images is shown in Figure 4. The volunteer wore striped socks so that we could extract more texture features and achieve a better reconstruction effect. On the one hand, the magazine was used to distinguish from the surrounding environment; on the other hand, it was used as a calibration object to facilitate the later measurement. To ensure more overlap in the images, we used the Free Studio to turn the video into images.

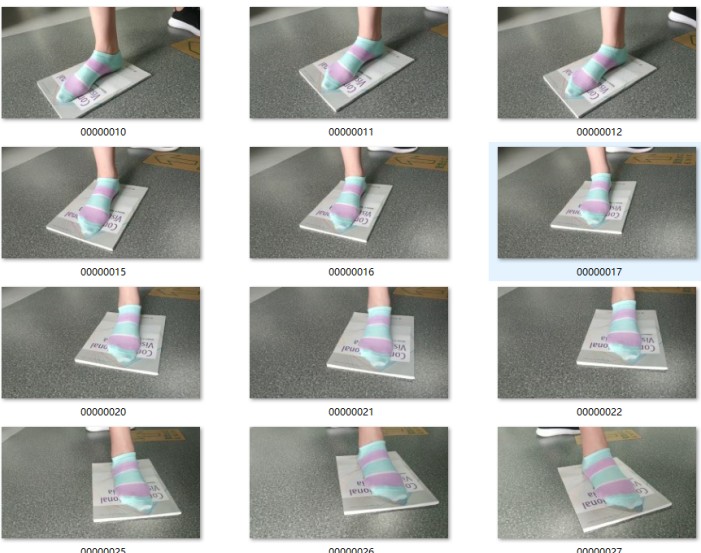

**Figure 4.** The part of the foot images.

### 4.2. 3D Foot Reconstruction in Visual SFM

The Visual SFM was used for 3D reconstruction. First, we imported 84 images, then SIFT feature points were detected and matched, and the RANSAC algorithm was used to eliminate mismatched feature points. Second, the feature points were estimated for 3D coordinates and camera parameters, and the Bundle Adjustment was used for parameter optimization to obtain a sparse point cloud. The sparse model is shown in Figure 5. The color polyhedron on the top is the camera position, and the bottom is the sparse point cloud of the foot model.

The dense point cloud reconstruction was based on PMVS. The sparse model is initialized feature matching; then, to ensure that each image block corresponds to at least one face, we adopted patch generation; finally, the patch filtering was adopted to further filter the target model to generate a dense model. We can see the dense model in the Meshlab.

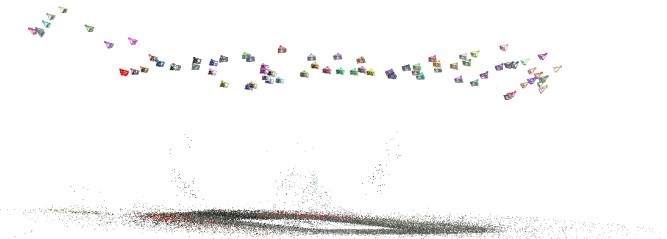

**Figure 5.** The sparse model in Visual SFM.

*4.3. Post-processing Process in Meshlab*

The point cloud was imported into the Meshlab. Step 1, we opened the Layer Dialog and checked whether the camera was loaded correctly. The Camera Scale Factor of this experiment was 0.0001, and the camera position was visible. Step 2, the visible sparse point cloud was hidden, and then the ply file generated by Visual SFM was imported; as shown in Figure 6, we can see many miscellaneous points.

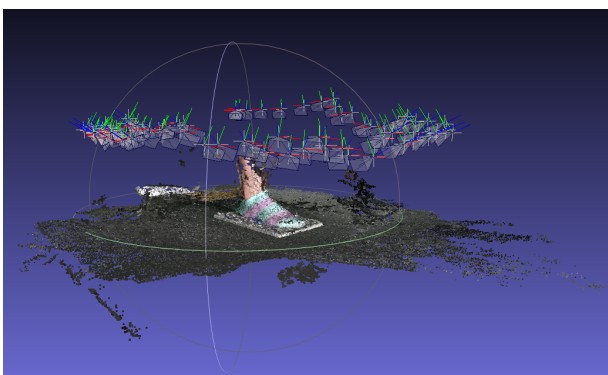

**Figure 6.** The dense model in Meshlab.

To improve the accuracy of reconstruction, step 3 selected and removed the miscellaneous points manually. Step 4 was to use the Poisson Surface Reconstruction algorithm to generate a polygonal mesh surface from dense point clouds. The Octree Depth parameter here controlled the mesh details. The larger the value, the more abundant the details are, but it also takes up more memory and runs slower. The use of the Poisson surface reconstruction algorithm will generate a "bubble" wrapping all scene objects in it, the model was closed, so it could later enclose other miscellaneous points, resulting in model redundancy. Therefore, in step 5, we needed to set a threshold to delete the redundant parts. Due to the texture, the processing requirements of grid model must be manifold, therefore we needed to remove the non-manifold edges in step 6 (that is any shared by the many faces of edge), step 7 was parameterized (according to the camera projection to create the UV mapping), and the texture projection was finally saved and exported; the export file can choose for the ply of the stl file, as well as the obj file, and so forth. The last 3D foot model is shown in Figure 7.

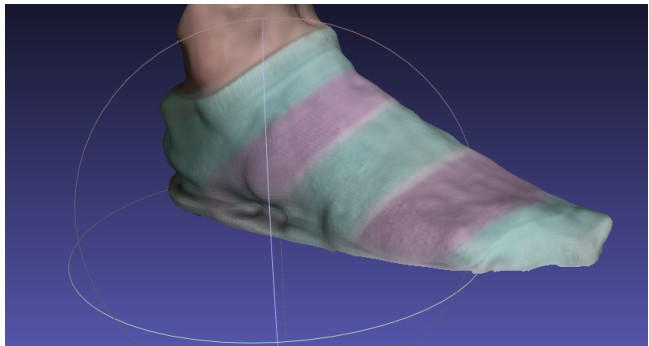

**Figure 7.** The 3D reconstruction foot model.

*4.4. The 3D Foot Model Measurement*

After obtaining the 3D foot model, we could use the measurement tool provided by Meshlab to obtain the key parameters of the foot model as grid units. In this paper, the Ground Control Point (GCP) method was used to convert the coordinates of the point cloud model into world coordinates by using the actual size. The foot, and the magazine of a known size as a calibration object, were modeled together, the relationship between the 3D coordinate system and the world coordinate system was obtained by the conversion relation between the actual size of the calibration object and in the 3D model.

We used the measurement tool provided by the Meshlab to measure the foot length and foot width as a grid unit, then converted them to millimeters in the real world using calibration objects. The parameters measured by hand were used as calibration values, and the foot data obtained by foot scanner as a contrast, to compute measurement errors.

We take the average value of the parameters in the unit of grid in the 3D coordinate system by multiple measurements—the foot length is 0.892023, and the foot width is 0.307358, and the magazine width is 0.7550 grid. Previously, it was known that the size of the magazine width is 210 mm in the actual world, so we can calculate the foot parameters as follows:

The left of the equal sign is the object size in the actual world, and the right of the equal sign is the 3D model size in the Meshlab:

$$\frac{\text{The foot width (object size)}}{\text{The magazine width (object size)}} = \frac{\text{The foot width (3D model size)}}{\text{The magazine width (3D model size)}} \tag{11}$$

In our experiment, the parameters above the formula are:

$$\frac{\text{The foot length (object size)}}{210 \text{ mm}} = \frac{0.892023 \text{ grid}}{0.7550.\text{grid}} \tag{12}$$

So we can find that the foot width is 85.49 mm, and we can also find that the foot width is 248.11 mm in the same way.

In this paper, the AIRAJ digital caliper with an accuracy of 0.01 mm was used. The parameters measured by this digital caliper were used as the calibration value, and we obtained the foot length and foot width were 249.20 mm and 84.42 mm, respectively. In this paper, the santi-foot scanner is used, as shown in Figure 1c. The right foot of the volunteer was scanned for comparison, and the length and width of the foot are 248.50 mm and 84.90 mm, respectively. The results are compared as follows (Table 1).

**Table 1.** The results of different measurements.

| Measurement Category | Foot Length/Error (mm) | Foot Width/Error (mm) |
|---|---|---|
| Handcraft | 249.20 | 84.42 |
| Foot-scanner | 248.50/(0.70) | 84.90/(0.48) |
| 3D foot recostruction | 248.11/(1.09) | 85.49/(1.07) |

*4.5. The Experimental Results*

As can be seen from Table 1, the experimental error of foot length and foot width of the foot scanner is 0.70 mm and 0.48 mm, respectively. The experimental error of foot length and foot width of the 3D foot reconstruction method used in this paper is 1.09 mm and 1.07 mm, respectively. It can be seen that the experimental error of the 3D foot reconstruction method is still slightly larger than that of the foot scanner. However, the experimental error of the 3D foot reconstruction method is around 1 mm, which is tolerable for applications such as shoe tree customization. The method is also not limited by location and equipment, with high flexibility and convenience.

In the process of preparing the foot data, we firstly used the camera of the smartphone to shoot around the foot at an angle of 45 degrees and 90 degrees from the magazine to obtain different height information and more than 60 photos were obtained. However, in the process of reconstruction by using Visual SFM, we found that only just over 20 photos were successfully matched in the feature detection and matching stage, which resulted in a poor result of the 3D foot model. We tried to increase the photos by reducing the distance when shooting, but the results did not improve. We guessed that the movement distance during manual shooting could not be controlled, resulting in a low correlation between photos. Therefore, we tried to use mobile phones to shoot video and cut the videos into photos through software. Finally, it was verified that the reconstruction effect of photos captured by video was better.

The results obtained by the 3D reconstruction method based on the foot images were compared with the results obtained by the 3D foot-scanner, as shown in Figure 8. The model has a little geometric distortion in the heel, and the prominent features of the toes and soles are relatively complete; the expected requirements are met. Some vacancies appear on the medial side of the arch and the heel of the foot. The reason is that due to the height of the camera and the angle of shooting, there are some shooting corners that cause the above part to fail to reach enough point cloud data, which will affect the final modeling results. It can be improved by adjusting the shooting angle, replacing the sole reference, resetting the target position, and improving the dense reconstruction method. At the same time, the result can only show the shape and appearance of the foot's shape. It is necessary to calculate the length of the target object and other parameters by comparison with the reference object. There may be some experimental error in the process. These errors can also be reduced as much as possible by the error compensation method proposed by Zhen S. et al. [30].

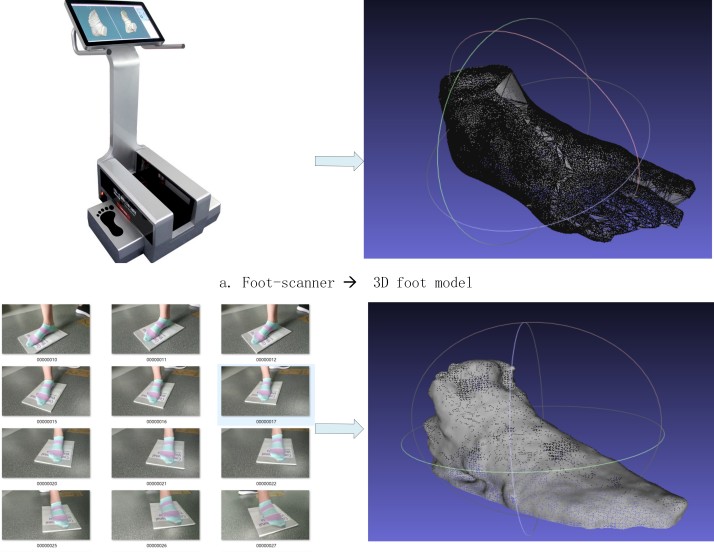

a. Foot-scanner → 3D foot model

b. Multi-view images → 3D foot model

**Figure 8.** 3D foot model comparison of foot-scanner and Multi-images.

The advantage of the 3D foot reconstruction method based on mobile phone photography is that the operation is simple, convenient, low cost and easy to spread, and the user can complete the foot reconstruction work at home by using a mobile phone. We can use the mobile phone as the scene information acquisition device. The acquisition time is relatively short and free, which can quickly complete the information acquisition process for the target object. However, the acquisition time of laser scanning and other methods is relatively long, and it is generally determined by the surface area of the measured target and the sampling rate of the measured space, which results in the measurement of the 3D geometric structure unsuitable for the biological shape. The methods used in this paper also have some shortcomings. For example, the density of model sampling points obtained directly is limited by camera resolution and object characteristics, which cannot achieve the same sampling density as laser scanning or measurement methods based on structural fringe grating. Secondly, feature matching is an important problem to be solved. Due to the image noise, camera distortion, illumination difference and the projective space itself, it is very difficult to automatically match features, and tedious interactive correction is usually required to obtain a more accurate model.

## 5. Conclusions

After obtaining the 3D foot model, the user can adjust the selected style of shoe tree and make the feature points, walking and other parts on the shoe tree more comfortable. After the adjustment, the factory can hit the entity shoe tree, then entity shoe tree to footwear processing workshops, producing shoes according to a user's foot type personalization. This method can solve the problem that many customers cannot buy suitable shoes.

As one of the applications of the foot model, a shoe tree was built according to the foot model, and different shoe models were designed based on the shoe tree model. It can realize the whole process of digitization and automation [31]. A large number of consumer foot data can be used to build a big enterprise foot database, helping designers to design more personalized and comfortable shoes. At the same time, it can also be used to compare new products with original design data, detect processing accuracy, improve production efficiency and ensure product quality.

The 3D model of the foot generated by this method is widely used; it can be used not only for high-end personalized customization, but also for medical and other situations. Here are a few examples:

- Orthopedic shoes custom: Collecting the patient's foot 3D model, producing the orthopedic shoes according to the patient's foot 3D model. At the same time, the orthopedic center can compare the foot model of the patients in different periods and observe the correction.
- Foot database: The data collection terminal obtains 3D foot data of the consumer with the permission of the consumer, and continuously sends the 3D foot data of the brand audience to the background database. It can easily establish an informative and comprehensive 3D foot database;
- Prosthetic production: Disabled patients have limited mobility, and can use the 3D reconstruction technology based on mobile phone photography to scan the patient's other healthy foot at home, and then get a complete 3D model of the foot quickly. Then we can produce prostheses and also make sure that the two feet are exactly the same;
- Medical diagnosis: Quickly collect 3D data of the patient's foot, and the podiatrist can accurately determine the shape of the patient's foot disease based on the collected 3D data, so as to adopt a more scientific and more effective treatment method;
- Surgical assistance: Using the collected 3D data of the patient's foot, through careful observation of the patient's foot in the computer, the best surgical plan to ensure the success rate of the operation can be designed;

In addition, the method can also be widely used in other areas to obtain 3D big data of the human body, such as smart wearable devices, medical treatment and health, and so

forth. Compared to the foot scanner, this method can reduce the costs of data collection and it does not require professional equipment so it is not limited by the venue; it is also flexible with high portability and good practicability. This method is also suitable for 3D reconstruction of multiple parts in multiple scenes.

Next, we will further improve the accuracy of the model by improving the dense reconstruction method and add the 3D model edge line extraction method to extract more foot shape parameters. We also plan to develop an application; this method could be very economically effective for manufacturing and healthcare. Nowadays, the 3D foot scanner can be divided into two types: handheld scanner and stationary scanner. A handheld 3D scanner is flexible, convenient and versatile, suitable for scanning complex structure multi-curved objects. The stationary 3D scanner has high accuracy, which is not affected by object color and light. The price of the scanner varies from 800 \$–30,000 \$, depending on its use and accuracy, which is expensive for customers. Our application, however, is low-cost and highly flexible; we use the mobile phone as a tool to obtain images, then the reconstruction process is deployed on the cloud. The user can upload images to the cloud, the cloud will start to deploy the reconstruction process after receiving the images, then the foot 3D model and key parameters will be sent to the user's phone; using only a mobile phone, users can obtain their foot 3D model. The costs include the software license and the human labour. The Visual SFM and Meshlab are open source software; we considered obtaining the software license for cooperation. The cost of the human labor included data acquisition and post-processing after obtaining the 3D foot model. In the future, data acquisition will be finished by the user, and, for the post-processing, we will consider using code to realize automated processing.

**Author Contributions:** G.X. and H.W. provided research idea and guidance, research fund and support; L.N. prepared the original draft and finish the experiment; C.G. and X.C. gave some experiment guidance and help, G.X. and X.S. reviewed and edited this paper as well as provided additional information. All authors have read and agreed to the published version of the manuscript.

**Funding:** This work was supported by the National Key Research and Development Program of China (2018YFB1702701), the National Natural Science Foundation of China (61773381, 61773382, U1909218), and Chinese Hunan's Science and Technology Project (2018GK1040).

**Institutional Review Board Statement:** Not applicable.

**Informed Consent Statement:** Not applicable.

**Data Availability Statement:** The data presented in this study are available on request from the corresponding author.

**Conflicts of Interest:** The authors declare no conflict of interest.

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
