# Peer review of "3D Foot Reconstruction Based on Mobile Phone Photographing"

_applsci, doi:10.3390/app11094040_

Round 1
Reviewer 1 Report
Tha authors have to some degree followed the recommendations of the reviewers. I recommend publication.
Reviewer 2 Report
Thank you for improving your work.
This manuscript is a resubmission of an earlier submission. The following is a list of the peer review reports and author responses from that submission.
Round 1
Reviewer 1 Report
The article presents a method of creating a 3D foot model with the use of smartphone shooting. The method uses SfM (Structure from Motion) algorithm, PMVS (Patch-based Multi View System) system, and finally Meshlab software for postprocessing. Meshlab is used to manually correct the 3D model and make final measurements. The method is an alternative to using 3D foot scanner. It also provides sufficient modeling accuracy compared to manual measurements and a 3D scanner. The method is one of the tools of Industry 4.0.
The authors suggest (only) that the method presented in the article is cheaper than using a 3D scanner. The method can be used to propose or produce customized shoes (including for rehabilitation and defect correction purposes).
The following questions / suggestions may help improve the quality of the article:
Q1. As I guess, the magazine was the calibration plate (section 4.4). Pls. explain the calibration process/tool at the beginning of section 4. And maybe extend experiment equipment and environmental requirements. Add the calibration plate (the magazine) as a needed device. I also suggest will extend method by adding the task of measuring the size of the calibration plate.
Q2. Calibration plate - the magazine – can has inaccurate dimensions and bends (does not form a plane) when the foot is placed. Both of these properties can introduce errors in the calibration and hence the result. Appropriate research and analysis could be carried out.
Q3. The proposed method uses generally available tools (smartphone and magazine), but also uses sophisticated software tools, as well as labor-intensive manual post-processing. It would be good to analyze the economic effectiveness of the method in relation to the use of a 3D scanner. The analysis should take into account the cost of the scanner, the software license and the cost of human labor during model building and others.
Small mistakes would be good to correct, for example (lines 203: [R|t]| ).
Author Response
We appreciate your warm work earnestly. We have made the modifications accordingly which are highlighted in yellow or deleted in strikeout in the revised paper, and the response for the question in the attchment. I hope that the correction will meet with approval.

Reviewer 2 Report
The manuscript presents an interesting application of 3D reconstruction from multiview acquisitions. For the proposed goal, (foot measurement), the combination of used tools provides a very convincing alternative to very expensive laser scanning technique, or the very crude manual measurement. The paper demonstrates very clearly that a robust, cheap, yet accurate reconstruction can be obtained. As such the topic of the paper is very well focused and of wide interest as it may be extended to a large variety of applications.
However, the presentation is not very satisfactory.
A large body of the presentation is devoted to a projective model where the author’s contribution is simply inexistent as they use a software “Visual SFM” without appropriate reference to the author:
WU, Changchang. Towards linear-time incremental structure from motion. In : 2013 International Conference on 3D Vision-3DV 2013. IEEE, 2013. p. 127-134.
This software encodes a number of the tools that are used, including the bundle adjustment. Thus the entire Section 3, pp. 3-8, (half the paper length) which is awkwardly copied from Zhang [19], or encoded in Visual SFM, … should be removed as it is never used explicitly in the paper. The authors are simply users of the code.
Rather the authors should concentrate on their own contribution: As users of the code, did they face any difficulty that required a special treatment? did they develop anything macro, or script to make it more efficient? In the details of the method, the authors chose striped socks. In fact a striped pattern is much more difficult than any other more complex design simply because only the direction perpendicular to the stripe can be used as a reference whereas along the stripe, the degeneracy is high. On this point a comment would be welcome.
The authors are also using Meshlab in order to construct a CAD model from the dense cloud of point. Again, as users, the only points that deserve comments are the new difficulties, new ideas that the authors proposed to meet their goal.
Finally, the author aimed at a quantitative measurement of the foot. And length and width are given with 1/100 mm accuracy, although the observed foot is measured with socks on, and twice the thickness of the socks is presumably in the mm range. Limiting the comparison to only two distances whereas two 3D models are available is not a dense set of quantitative comparison… Thus even is one is ready to believe the plausibility of the result, a more systematic and cleaner comparison is really needed.
Additionally, introduction, discussion and conclusion are very redundant, and avoiding repetition will make the paper more attractive.
As a conclusion, the topic certainly deserves publication, but the paper should be rewritten entirely, focusing on the authors’ own contributions
Author Response

(The authors gave the same response as above.)

Round 2
Reviewer 2 Report
The authors have made an effort to follow my previous suggestions, and have shortened the paper. I consider that now the manuscript can be published in the present form (with minor language corrections, especially in the added text of this revision).
Author Response
We appreciate the Reviewer’s and the Assistant Editor for their warm work earnestly. We have made the modifications accordingly which are highlighted in yellow in the revised paper, and also recorded the modification points as follows. We hope that the correction will meet with approval.
